# Effects of Matched and Mismatched Visual Flow and Gait Speeds on Human Electrocortical Spectral Power

**DOI:** 10.3390/brainsci15050531

**Published:** 2025-05-21

**Authors:** Yu-Po Cheng, Andrew D. Nordin

**Affiliations:** 1Texas A&M Institute for Neuroscience, Texas A&M University, College Station, TX 77840, USA; 2Department of Biomedical Engineering, University of Houston, Houston, TX 77204, USA; adnordin@uh.edu

**Keywords:** EEG, visual flow, gait speed, virtual reality, spectral power

## Abstract

**Background/Objectives**: Visuomotor integration relies on synchronized proprioceptive and visual feedback during visually guided locomotion. How the human brain processes unimodal or asynchronous multimodal inputs during locomotion is unclear. **Methods**: Using high-density mobile electroencephalography (EEG) and motion capture in a virtual reality environment, we investigated electrocortical responses during altered treadmill gait speeds (0.5 and 1.5 m/s) and visual flow speeds (0.5×, 1×, and 1.5× gait speed) among 13 healthy human subjects. Experimental conditions included passive viewing of a moving virtual environment, walking in a stationary virtual environment, and walking in a moving environment with synchronous and asynchronous visual flow. **Results**: At faster gait speed, we identified reduced premotor, sensorimotor, and visual electrocortical beta-band spectral power (13–30 Hz) and greater premotor cortex theta power (4–8 Hz). At faster visual flow speeds, we identified reduced sensorimotor electrocortical beta-band spectral power, reduced alpha (8–13 Hz) and beta power, and greater gamma-band power (30–50 Hz) from the visual cortex. During visual flow and gait speed mismatches, sensorimotor and parietal alpha- and beta-band electrocortical spectral power decreased at faster gait speed. During treadmill walking at 1.5 m/s, parietal electrocortical spectral power increased when visual flow exceeded gait speed. **Conclusions**: Electrical brain dynamics during human gait identified distinct neural circuits for integrating kinesthetic and visual information during visuomotor conflicts, gated by the parietal cortex.

## 1. Introduction

During daily life, vision and kinesthesia are seamlessly integrated by visuomotor brain regions to facilitate visually guided locomotion [1]. Visual motion information first reaches the striate cortex (V1) then passes through the extrastriate (V2, V3) and occipitotemporal regions (V5/V5a) prior to the parietal cortex for visuospatial processing [2], within remarkably short durations under 150 ms [3]. Neuronal responses from the primary visual cortex of rodents are modulated by eye movements and behavioral states and are driven by locomotion and mismatches between received and expected visual inputs [4,5,6]. Mice raised in dark environments have shown neuronal responses less correlated with visual flow when exposed to random visuomotor coupling compared to those with normal visuomotor coupling [7]. Despite known differences between rodents and primates regarding eye positions, visual acuity, and neural circuits, peripheral visual processing between rodents and primates is believed to be similar, providing a suitable model for studying the neural basis of optic flow processing [8]. Rodent visual processing models during locomotion afford detailed dissections of mammalian neural circuits that await verification/comparison in non-human primates and humans [9].

Visuospatial processing has been studied in humans to better understand cognitive functions and gaze behaviors from immobile participants navigating virtual environments via keyboard or controller [10,11,12] and while viewing first-person video during gait [13]. During visual processing of optic flow [14,15,16], event-related potentials from EEG recordings over occipital and parietal areas have shown reduced N2 peak amplitudes and greater latencies at faster driving speeds [17] but similar latencies among walking, jogging, and cycling speeds [16]. Distance estimation during walking has shown that visual flow faster than walking speed led to higher variability in judgment error in healthy controls compared to Parkinson’s patients when speeds did not match [18].

Mobile electroencephalography (EEG) provides a valuable tool for investigating brain activity during human locomotion [19,20,21]. Advancements in mobile EEG hardware and signal processing [20,21,22,23] have enabled the identification of electrocortical spectral power fluctuations throughout the gait cycle [20] and reduced alpha- and beta-band spectral power at faster gait speeds [21]. Sustained reductions in alpha- and beta-band electrocortical spectral power are established biomarkers for increased cortical engagement, reflecting visuospatial processing from the visual and parietal cortices [24,25,26], sensorimotor processing in the primary motor and somatosensory cortices [21,23,27,28], and error monitoring from the frontal and anterior cingulate cortices [29,30]. During spatial navigation with matched and mismatched visual, kinesthetic, and vestibular information, vestibular perturbations led to a greater alpha power reduction from parietal, occipital, and temporal areas and increased spectral power from frontal cortical areas compared to matched conditions [31]. Despite prior research investigating visuospatial processing during gait, a gap remains in our understanding of how asynchronous visual and kinesthetic feedback influences brain activity in ambulatory humans.

Our aim was to identify electrocortical spectral power dynamics during treadmill walking with matched and mismatched visual and kinesthetic feedback. To achieve this, we manipulated treadmill walking speed and visual flow using a projected virtual reality (VR) environment. We hypothesized that passively viewing a moving VR environment while the body remained stationary would reduce visual and parietal alpha- and beta-band electrocortical spectral power at faster visual flow speeds due to greater visuospatial processing. We further hypothesized that during active walking, sensorimotor alpha- and beta-band spectral power would decrease at faster gait speeds, indicative of greater sensorimotor processing. Compared to walking with matched visual flow and gait speeds, we anticipated that mismatched visual flow and gait speeds would identify reduced alpha- and beta-band spectral power from the visual and parietal cortices along with the frontal and anterior cingulate cortices due to greater cognitive error processing from conflicting visual and proprioceptive feedback. Deciphering electrical brain dynamics during multisensory integration can help to identify neural pathways impaired by neurological disease or disorder and provides the basis for developing locomotor brain–computer interfaces.

## 2. Materials and Methods

A total of 13 healthy human subjects (7F, 6M, mean age 28.6 ± 10.3 years) participated in the study. The experimental protocol received institutional review board approval, and each participant provided written informed consent prior to participation.

We used a 10-camera motion capture system (VICON, Oxford, UK) to record lower-limb kinematics (100 Hz), and ground reaction forces were recorded (at 1000 Hz) from a split-belt force-measuring treadmill (M-Gait, DIH, Amsterdam, The Netherlands). To record mobile high-density EEG, we used a 64-channel system (LiveAmp, BrainProducts, Gilching, Germany) synchronized to the motion capture and force plate system. Virtual environments were generated by D-Flow software (version 3.34.3, DIH, Amsterdam, The Netherlands) and projected onto a screen at the front of the treadmill and the treadmill belts by two separate projectors. The VR content was presented as a virtual camera that moved along a straight path in a fixed virtual environment, where the projection onto the screen contained radial optic flow and the projection onto the treadmill belts presented translational optic flow, replicating real-world walking experiences when human subjects walk down a straight path. Visual flow did not account for the participant’s motion.

Participants completed 14 total conditions in 3 visual flow and gait speed combinations in a pseudorandom order: during quiet standing while viewing a stationary or moving projected virtual reality scene (Figure 1A; 0 m/s, 0.25 m/s, 0.5 m/s, 0.75 m/s, 1.5 m/s, and 2.25 m/s), during treadmill walking without visual flow (Figure 1B; 0.5 and 1.5 m/s), and during conditions with matched and mismatched gait speed and visual flow speeds (Figure 1C; walking at 0.5 and 1.5 m/s, with visual flow speeds 0.5× 1.0×, and 1.5× gait speed). Visual flow speed (VR environment moving speed) and gait speed (treadmill belt speed) were adjusted using D-Flow software.

We used an adapted data pre-processing pipeline to remove noise from our EEG recordings [21]. A 1 Hz high-pass filter and robust reference was applied [32], followed by sliding window spectral cleaning using principal component analysis to remove transient large-amplitude fluctuations [21] and canonical correlation analysis to remove electrical and muscle activity artifacts [33,34]. After concatenating all pre-processed data among conditions, we rejected remaining noisy channels and down-sampled the data to 256 Hz before applying an Adaptive Mixture Independent Component Analysis in EEGLAB [35]. Independent components were labeled for likelihood of brain, muscle, eye, or noisy signals [36], where components with 90% or higher chance to be non-brain signals were rejected. Independent components were modeled as equivalent current dipoles using DIPFIT in EEGLAB, retaining components with residual variance below 15%. We separated the continuous EEG data into the experimental conditions and calculated mean power spectral density, removing aperiodic (1/f) noise from the spectral data [37]. Based on an independent component power spectrum, scalp map, dipole location, and dipole orientation, we applied K-means clustering (k = 15) to evaluate condition comparisons, averaging independent components within subjects for each cluster. The resultant cluster locations were consistent during repeated clustering, showing an average silhouette score of 0.53, indicating reasonable to strong quality.

We performed gait event detection using the vertical position of the heel and toe markers from the motion capture data. The heel strike events were defined as the local minimum of the heel marker’s z-axis position. The toe off events were defined as the nearest zero-crossing point in the toe marker’s z-axis velocity before the foot left the ground. Gait timing was defined using the duration between relevant footfall events: heel strike to ipsilateral heel strike for stride time, heel strike to ipsilateral toe off for stance time, and heel strike to contralateral heel strike for step time.

To analyze the effects of gait speed and visual flow speed on EEG spectral power, we performed separate Friedman tests within each spectral band: delta (2–4 Hz), theta (4–8 Hz), alpha (8–13 Hz), beta (13–30 Hz), and gamma (30–50 Hz). Statistically significant main effects were followed up by pairwise Wilcoxon signed rank tests with Bonferroni corrections. One-way analysis of variance was used to compare gait timing and variability in stride time, stance time, and step time from both left and right limbs. Pairwise comparisons were corrected using the Bonferroni method.

## 3. Results

We identified dipole clusters from the premotor, sensorimotor, parietal, and visual cortices (Figure 2) and investigated the effects of gait speed and visual flow speed on spectral power from these regions.

When participants walked in a stationary virtual reality environment at 1.5 m/s compared to 0.5 m/s, beta-band spectral power decreased from the premotor (*p* = 0.031), sensorimotor (*p* = 0.0039), and visual cortices (*p* = 0.012) (Figure 3). Premotor electrocortical theta power increased during walking at 1.5 m/s compared to 0.5 m/s (*p* = 0.031). The parietal cortex did not show spectral power changes in response to changes in gait speed (Appendix A).

When participants stood motionless while viewing a moving virtual reality environment, faster visual flow speeds showed reduced visual electrocortical alpha *(X*^2^_(4)_ = 21.05, *p* < 0.001) and beta spectral power (*X*^2^_(4)_ = 19.69, *p* < 0.001) and greater gamma power (*X*^2^_(4)_ = 11.94, *p* = 0.018) (Figure 4). Sensorimotor beta-band spectral power decreased at faster visual flow speed (*X*^2^_(4)_ = 11.28, *p* = 0.024). Premotor cortex showed a statistically significant main effect for reduced beta-band spectral power at faster visual flow speeds (*X*^2^_(4)_ = 10.80, *p* = 0.029), but pairwise comparisons failed to show differences (Appendix A). Spectral power from the parietal cortex did not show differences in response to changes in visual flow speed when standing motionless (Appendix A).

Compared to standing motionless with fixed visual flow speed (0.75 m/s), when participants walked slower (0.5 m/s) or faster (1.5 m/s) than visual flow speed, electrocortical spectral power decreased at faster gait speed within alpha and beta bands from the sensorimotor (alpha: *X*^2^_(2)_ = 14.60, *p* < 0.001; beta: *X*^2^_(2)_ = 18.20, *p* < 0.001) and parietal cortices (alpha: *X*^2^_(2)_ = 9.75, *p* = 0.0076; beta: *X*^2^_(2)_ = 9.75, *p* = 0.0076) (Figure 5). The premotor cortex showed a statistically significant main effect for reduced alpha- (*X*^2^_(2)_ = 8.33, *p* = 0.016) and beta-band spectral power (*X*^2^_(2)_ = 6.33, *p* = 0.042) at faster visual flow speeds, but pairwise comparisons failed to show differences (Appendix A). Spectral power from the visual cortex did not change during gait speed alterations at fixed visual flow speed (Appendix A).

Mismatches between gait and visual flow speeds did not alter electrocortical spectral power during treadmill walking at 0.5 m/s (Appendix A). During treadmill walking at 1.5 m/s, mismatches between gait and visual flow speed altered alpha spectral power from the parietal cortex (*X*^2^_(2)_ = 6.8, *p* = 0.034). Compared to matched visual flow and gait speeds, alpha-band spectral power from the parietal cortex increased (*p* = 0.0078) when visual flow exceeded gait speed (Figure 6). The visual cortex showed a significant main effect (*X*^2^_(2)_ = 9.7, *p* < 0.01) of visual flow speed on beta-band spectral power, but pairwise comparisons failed to show differences (Appendix A). No changes were identified from the premotor and sensorimotor cortices (Appendix A).

Mismatched visual flow and gait speed did not alter gait timing or the timing variability of strides, stances, and steps compared to matched conditions (Figure 7).

## 4. Discussion

We aimed to identify electrocortical spectral power dynamics during treadmill walking with matched and mismatched visual and proprioceptive feedback. By recording high-density EEG and three-dimensional motion capture during treadmill walking in a projected VR environment, we identified changes in premotor, sensorimotor, parietal, and visual electrocortical spectral power that partially confirmed our hypotheses. At faster gait speed, we identified reduced premotor, sensorimotor, and visual electrocortical beta-band spectral power (13–30 Hz) and greater premotor cortex theta power (4–8 Hz). At faster visual flow speeds, we identified reduced sensorimotor electrocortical beta-band spectral power, reduced alpha (8–13 Hz) and beta power, and greater gamma-band power (30–50 Hz) from the visual cortex. During visual flow and gait speed mismatches, sensorimotor and parietal alpha- and beta-band electrocortical spectral power decreased at faster gait speed. During treadmill walking at 1.5 m/s, parietal electrocortical spectral power increased when visual flow exceeded gait speed.

### 4.1. Effects of Gait Speed and Visual Flow Speed

Reduced alpha- and beta-band electrocortical spectral power has been consistently reported as a correlate of greater somatosensory processing from the sensorimotor cortex [21,23,38] and greater visual processing from the visual cortex [31,39], reflecting increased cortical excitability and neuronal firing rates. When participants walked in a stationary visual environment, faster gait speed led to reduced sensorimotor electrocortical spectral power, in alignment with previous results [21]. Reduced spectral power from the premotor cortex has been reported in conditions requiring greater motor planning and gait modification [38,40]. We considered reduced beta-band spectral power from the visual cortex in relation to two possible mechanisms that increased visual cortex excitability: (1) movement-related modulation by the motor system and (2) greater variation in retinal images caused by head and limb movements at faster gait speeds. Studies in marmosets [41] and macaque monkeys [42] have shown that body movements have a lesser effect on visual electrocortical activity compared to rodents, indicating that changes in visual input at faster gait speeds, even while viewing a stationary scene, more likely caused our observed reduced beta-band spectral power from the visual cortex.

When participants stood still and viewed the moving scenery at varied visual flow speeds, alpha- and beta-band spectral power reduced at faster speeds from the visual cortex, indicating greater visual processing. Beta-band desynchronization has been previously identified during visual motion among seated adult subjects viewing optic flow [15]. We identified greater visual gamma-band spectral power at faster visual flow speeds. Greater visual gamma power often accompanies reduced alpha/beta power [43] and has been linked to the feedforward drive of sensory information that is thought to promote information transfer in human and non-human primates [44,45,46].

At faster visual flow speeds, we identified reduced sensorimotor beta-band spectral power. Optic flow plays a critical role in maintaining upright balance in young and older adults, showing reduced alpha-band spectral power from the sensorimotor area when viewing optic flow versus static stimuli [47]. Visual feedback can influence postural control and elicit body movements such as visually induced postural sway when standing and viewing periodically changing optic flow [48,49,50]. Postural control has also been attributed to increased theta-band power [51,52] and reduced beta-band power [52].

### 4.2. Effects of Mismatched Gait and Visual Flow Speeds

Locomotion creates visual flow in synchronization with self-movement, which is encoded by the visuomotor neural circuits [8]. Developmental experiences can modulate the encoding mechanism. Mice raised with random visuomotor coupling failed to develop typical interactions between locomotion-related and visual inputs in layer 2/3 of the primary visual cortex compare to those raised with normal visuomotor coupling [7]. Human infants with at least 3 weeks of crawling experience, compared to infants with no crawling experience, exhibited electrical brain responses to visual flow speed and direction similar to juveniles and adults [15,16]. Recent studies investigating sensorimotor mismatches in rodents [4,5,6] have identified around 10% of visual cortical neurons in L2/3 capable of detecting visual flow discrepancies from self-motion. In humans, reduced parietal and occipital alpha-band spectral power has been identified when changing walking direction, with stronger parietal alpha desynchronization during sensory mismatches [31].

When visual flow speed was fixed and gait speed was altered, sensorimotor alpha- and beta-band electrocortical spectral power reduced at faster gait speed, in agreement with prior studies [21,23,38]. The parietal cortex also showed reduced alpha- and beta-band power at faster gait speeds, indicative of greater visuospatial processing, which has been identified among younger and older adults [53].

When gait speed was fixed and visual flow speed was altered, electrocortical spectral power increased from the parietal cortex. Because of our study design, a greater range of visual flow speeds occurred during 1.5 m/s walking compared to 0.5 m/s (e.g., 1.5 × 1.5 m/s > 1.5 × 0.5 m/s), which could explain why only walking at 1.5 m/s with faster visual flow speed showed changes in parietal electrocortical spectral power. Human EEG studies that assessed event-related potentials during gait have also identified discrepancies between small [16] and large [17] speed ranges. In opposition to our hypothesized electrocortical spectral power reductions during sensorimotor conflicts, greater parietal spectral power could be due to reduced (1) visuospatial processing, (2) attention reallocation, or (3) predictive coding. Prior studies have focused on introducing sensorimotor conflicts in a small portion in the visual field on a computer display [40] or a transient need for updating spatial information based on changing direction [31] or entering a tunnel [54]. We speculate that the scale and duration of our gait and visual flow speed mismatches could have suppressed electrocortical responses during adaptation. Inhibition of the temporoparietal junction has been thought to prevent attention reallocation [55], indicative of focused attention [56]. Predictive coding is achieved by continuous comparison between predicted and perceived sensory information throughout hierarchical structures [57], which could be based on prior experiences or efferent copies [58]. Our observed spectral power increases could be indicative of absent prediction errors, suggesting the nervous system has adapted to the new relationship between visual flow and gait.

The presence of altered parietal electrocortical spectral power in response to gait and visual flow speed mismatches, but not to independent changes in gait or visual flow speed, identifies its unique role in visuospatial processing. Greater parietal desynchronization has been identified from humans moving through space and changing direction when passively moving on a cart compared to walking [31]. Reduced parietal alpha-band spectral power has also been observed during walking when human subjects viewed movement-related videos compared to a black screen and objects irrelevant to movement [40]. These findings suggest that the parietal cortex can discriminate the relevance of self-movement feedback.

We did not observe reduced occipital spectral power in response to gait and visual flow speed mismatches. Although previous animal research has suggested that neurons in rodent primary visual cortex layer 2/3 receive copies of motor output and can detect sensorimotor conflicts [7], our results did not directly support this at a macroscopic EEG recording scale that lacks the spatial resolution for targeting specific cortical layers. Species differences must also be acknowledged, where movement-related visual cortical responses were lesser in primates compared to rodents [41,42].

Gait timing and gait timing variability did not show changes in response to mismatched visual flow and gait speeds. Faster optic flow speed has been reported to decrease walking speed [59,60,61] and alter the transition speed from walking to running and from running to walking [59]. Comparisons between overground walking and walking on an omnidirectional treadmill in virtual reality have shown reduced gait speed and shorter step lengths due to the novelty of physically navigating a virtual environment [62]. Because gait speed was maintained by the treadmill belt in our study, gait timing was preserved despite eliciting changes in electrocortical spectral power. Our findings are congruent with prior kinematic results during treadmill walking at fixed speeds when stride time was not modulated by mismatched visual flow and gait speeds [63]. Although the visual and parietal cortices showed spectral power differences in response to mismatched visual flow and gait speed, electrocortical changes were independent of biomechanical outcomes. The ability to identify changes in sensory processing based on electrocortical spectral power in the absence of biomechanical adaptations is critical for better understanding multisensory integration during human locomotion.

## 5. Conclusions

We identified distinct electrocortical responses to changes in visual flow and gait speed and mismatches between visual flow and gait speed. The premotor, sensorimotor, and visual cortices showed changes in spectral power in response to changes in either gait speed or visual flow speed, and parietal electrocortical spectral power showed changes specifically during mismatches between gait and visual flow speeds. The critical role of the parietal cortex in processing visuomotor conflicts comes from its intricate network of multimodal inputs that connect to multiple parts of the motor system. Identifying electrocortical responses to sensory mismatches during gait highlights the sensitivity of mobile EEG in capturing neural processes that occur without observable behavioral adjustments. Better understanding of the neural dynamics underlying sensorimotor integration during gait can help identify sources and potential treatments for locomotor deficits. Our observed differential responses among brain regions during gait and visual flow speed adjustments can also be used to develop targeted VR-based gait neurorehabilitation and for feature selection for the development of assistive brain–computer interfaces for visually guided walking.

## Figures and Tables

**Figure 1 brainsci-15-00531-f001:**
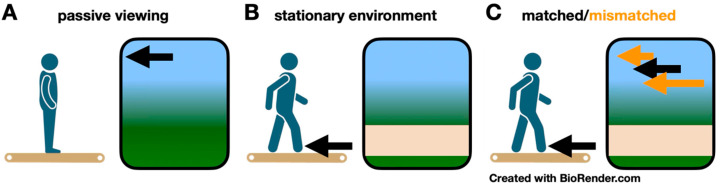
Visual flow and gait speed experimental conditions. (**A**) Passive viewing. Subjects stood motionless on the treadmill while virtually advancing through the environment at five visual flow speeds. The black arrow indicates the virtual environment moved at a fixed velocity. (**B**) Stationary environment. Subjects walked at two fixed gait speeds while viewing a stationary environment. The black arrow indicates the treadmill belts moved at a fixed velocity. (**C**) Matched/mismatched. Subjects walked at two fixed gait speeds, while visual flow matched gait speed, moved faster than gait speed, or moved slower than gait speed. The black arrows indicate the virtual environment and treadmill belts moved at the same velocity. The orange arrows indicate the virtual environment moved at a different velocity from the treadmill belts.

**Figure 2 brainsci-15-00531-f002:**
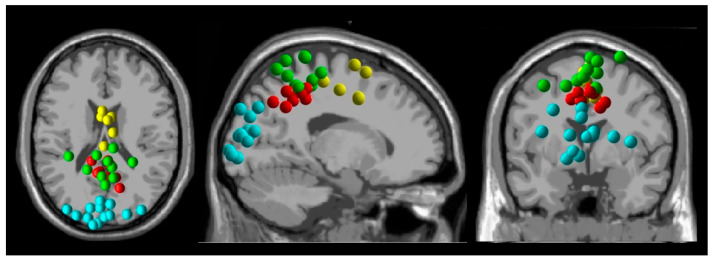
Electrocortical dipole cluster locations. Premotor (yellow), sensorimotor (green), parietal (red), and visual (cyan) cortices.

**Figure 3 brainsci-15-00531-f003:**
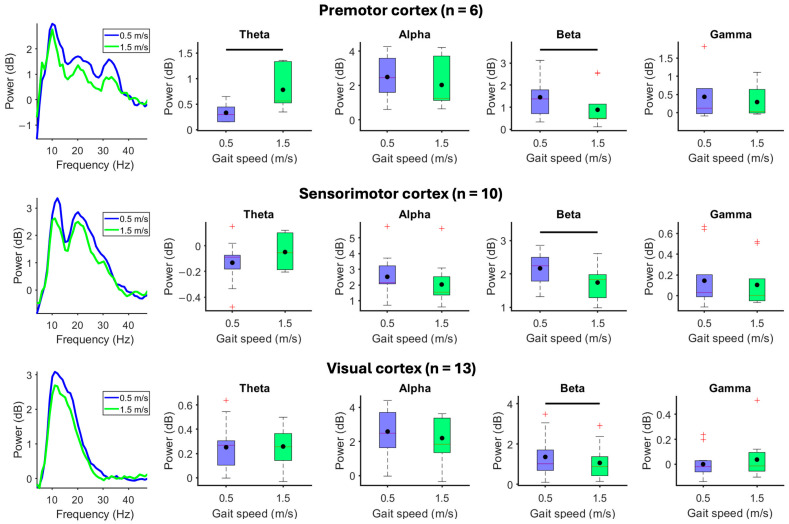
Effect of gait speed without visual flow. (**Left**) Mean spectral power in each gait speed condition. (**Right**) Mean spectral power boxplots aggregated within theta (4–8 Hz), alpha (8–13 Hz), beta (13–30 Hz), and gamma (30–50 Hz) bands when participants viewed a stationary virtual reality environment and walked at 0.5 or 1.5 m/s. Group means are black dots. Significant condition comparisons (black horizontal line, Wilcoxon signed rank test, *p* < 0.05). Red plus signs are outliers.

**Figure 4 brainsci-15-00531-f004:**
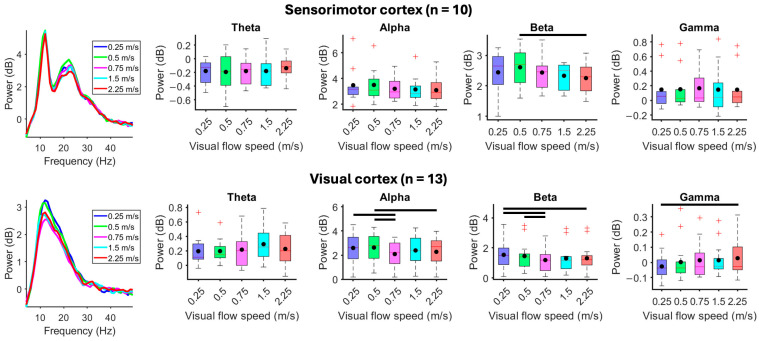
Effect of visual flow without walking. Subjects stood motionless on the treadmill and passively viewed the virtual reality environment moving at 0.25, 0.5, 0.75, 1.5, or 2.25 m/s. (**Left**) Mean spectral power during each visual flow speed condition. (**Right**) Mean spectral power boxplots aggregated within theta (4–8 Hz), alpha (8–13 Hz), beta (13–30 Hz), and gamma (30–50 Hz) bands. Group means are black dots. Significant condition comparisons (black horizontal lines, Wilcoxon signed rank test, *p* < 0.005). Red plus signs are outliers.

**Figure 5 brainsci-15-00531-f005:**
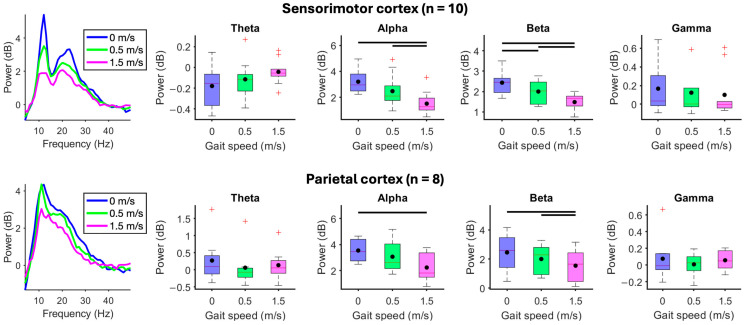
Effect of gait speed at fixed visual flow speed. Subjects viewed a virtual reality environment moving at 0.75 m/s while standing motionless (0 m/s) or walking at 0.5 or 1.5 m/s. (**Left**) Mean spectral power in each gait speed condition. (**Right**) Mean spectral power boxplots aggregated within theta (4–8 Hz), alpha (8–13 Hz), beta (13–30 Hz), and gamma (30–50 Hz) bands. Group means are black dots. Significant condition comparison (black horizontal line, Wilcoxon signed rank test, *p* < 0.017). Red plus signs are outliers.

**Figure 6 brainsci-15-00531-f006:**
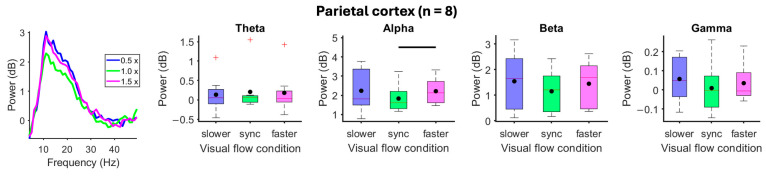
Effects of mismatched gait and visual flow speeds during 1.5 m/s treadmill walking. The virtual reality environment moved at 0.5×, 1×, or 1.5× gait speed. (**Left**) Mean spectral power during mismatched visual flow and gait speed. (**Right**) Mean spectral power boxplots aggregated within theta (4–8 Hz), alpha (8–13 Hz), beta (13–30 Hz), and gamma (30–50 Hz) bands. Group means are black dots. Significant condition comparisons (black horizontal line, Wilcoxon signed rank test, *p* < 0.017). Red plus signs are outliers.

**Figure 7 brainsci-15-00531-f007:**
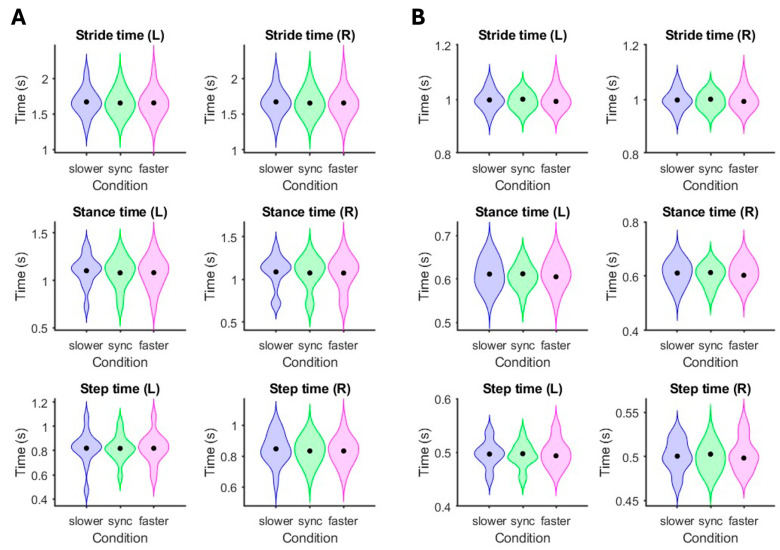
Effects of mismatched gait and visual flow speeds on gait timings. Violin plots showing stride time, stance time and step time while participants walked on a treadmill at (**A**) 0.5 m/s and (**B**) 1.5 m/s. Group means are indicated with black dots. The virtual reality environment moved at 0.5×, 1×, or 1.5× gait speed.

## Data Availability

The original contributions presented in this study are included in the article/Appendix A. Further inquiries can be directed to the corresponding author.

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
