# Peer review of "Effects of Matched and Mismatched Visual Flow and Gait Speeds on Human Electrocortical Spectral Power"

_brainsci, 2025, doi:10.3390/brainsci15050531_

Round 1

Reviewer 1 Report

Comments and Suggestions for Authors

The authors investigate the dynamics of electrocortical spectral power in different frequency ranges with matching and conflicting values of the speed of visual flow and the speed of walking on a treadmill in virtual reality. This is an extremely interesting study, since the issue of de-synchronizing the speed of movement of the image and the speed of movement is extremely relevant for the field of virtual reality. 
The authors used a 64-channel EEG and a motion capture system (VICON) as an objective data source. During the experiment, the indicators of brain activity were compared in a stationary state, without external influence of virtual reality and with varying degrees of synchronization of virtual images and real movement. 

After reading it, I had some questions, so the research is really interesting:
- what was the VICON motion capture system used for? I see only one S6 drawing in the accompanying materials. I think it's worth moving it to the main article in a separate paragraph and describing the specifics of movements with different combinations of speed of visual flow and real movement, as this is also a very important part of the study. Moreover, there is a part of this description in section 2, lines 115-120.
- do I understand correctly that the speed of the treadmill and the speed of the visual flow in the helmet were continuous and constant (taking into account the conditions of a particular experiment)? Was the content in the helmet a video recording or an interactive scene? I'll explain. When moving along the treadmill, in addition to the speed of the canvas itself, there is some component of a person's step during movement (local deceleration and acceleration), which in the case of an integrative scene can be taken into account since the virtual camera also moves behind the head. In the case of a video, this effect will not be present. 
- did the movement on the treadmill, electromagnetic interference from the treadmill's engines affect the EEG data? Because this is a fairly sensitive equipment and fast walking can have a negative impact by adding artifacts. In addition, the very movement of the head in the helmet can add artifacts.
- it would be interesting to see a correlation matrix (Heatmap) of input and output features, if possible. If there is no such possibility, then this is just a wish, not a requirement.

Author Response

Thank you for your recognition and insightful comments on our manuscript. Your comments regarding the details of study designs have greatly helped us improve the manuscript’s clarity and readability. We have carefully considered each of your suggestions and made edits to this paper.

Comment 1: What was the VICON motion capture system used for? I see only one S6 drawing in the accompanying materials. I think it's worth moving it to the main article in a separate paragraph and describing the specifics of movements with different combinations of speed of visual flow and real movement, as this is also a very important part of the study. Moreover, there is a part of this description in section 2, lines 115-120.
Response 1: Thank you for your question. We used the motion capture data to identify gait event timings using the heel and toe marker Z-coordinates and for synchronizing the EEG data based on the gait cycle. Based on your helpful suggestion, we have moved Supplementary Figure 6 to the Results section in the main text (lines 177-178, line 193), and included a relevant citation showing similar results (lines 286-291).

Comment 2: Do I understand correctly that the speed of the treadmill and the speed of the visual flow in the helmet were continuous and constant (taking into account the conditions of a particular experiment)? Was the content in the helmet a video recording or an interactive scene? I'll explain. When moving along the treadmill, in addition to the speed of the canvas itself, there is some component of a person's step during movement (local deceleration and acceleration), which in the case of an integrative scene can be taken into account since the virtual camera also moves behind the head. In the case of a video, this effect will not be present. 
Response 2: Thank you for your question. During testing, the treadmill belts maintained constant velocity. The virtual reality content was not presented by a VR helmet or VR goggles, but projected onto a screen at the front of the treadmill and on the treadmill belt by 2 projectors, respectively. The VR content was presented as a virtual camera moving straight along a fixed virtual environment at a matched or mismatched speed to the treadmill belts. The movement of a participant did not contribute to controlling the VR content and can be viewed as a video recording for each condition (detailed in lines 92-99).

Comment 3: Did the movement on the treadmill, electromagnetic interference from the treadmill's engines affect the EEG data? Because this is a fairly sensitive equipment and fast walking can have a negative impact by adding artifacts. In addition, the very movement of the head in the helmet can add artifacts.
Response 3: We greatly appreciate your concern regarding this data quality issue. To address the locomotor-related noise from motion, muscle, and electrical artifacts, we applied an adapted mobile EEG data processing pipeline that has been rigorously validated and used for studying human electrical brain dynamics during gait. We used an elastic EEG cap with the EEG electrodes firmly secured to the head and the EEG cables were secured on a safety vest to reduce cable sway.

  • Symeonidou, E. R., Nordin, A. D., Hairston, W. D., & Ferris, D. P. (2018). Effects of cable sway, electrode surface area, and electrode mass on electroencephalography signal quality during motion.Sensors18(4), 1073.

  • Nordin, A. D., Hairston, W. D., & Ferris, D. P. (2018). Dual-electrode motion artifact cancellation for mobile electroencephalography.Journal of neural engineering15(5), 056024.

  • Nordin, A. D., Hairston, W. D., & Ferris, D. P. (2019). Human electrocortical dynamics while stepping over obstacles.Scientific reports9(1), 4693.

  • Nordin, A. D., Hairston, W. D., & Ferris, D. P. (2019). Faster gait speeds reduce alpha and beta EEG spectral power from human sensorimotor cortex. IEEE Transactions on Biomedical Engineering67(3), 842-853.

  • Richer, N., Downey, R. J., Hairston, W. D., Ferris, D. P., & Nordin, A. D. (2020). Motion and muscle artifact removal validation using an electrical head phantom, robotic motion platform, and dual layer mobile EEG.IEEE transactions on neural systems and rehabilitation engineering28(8), 1825-1835.

  • Song, S., & Nordin, A. D. (2025). Cortical processing and lower limb muscle activity increase during bodyweight supported treadmill locomotion underwater compared to on-land.IEEE Transactions on Neural Systems and Rehabilitation Engineering.

Comment 4: It would be interesting to see a correlation matrix (Heatmap) of input and output features, if possible. If there is no such possibility, then this is just a wish, not a requirement.
Response 4: Thank you very much for bringing this suggestion to our attention. We agree that the suggested information could be relevant, but it may be more suitable for a methods-related paper. Nevertheless, we are confident in our applied methods and results based on the application of established data analysis procedures in our paper. 

Reviewer 2 Report

Comments and Suggestions for Authors

This study investigates electrocortical dynamics during treadmill walking with matched and mismatched visual flow speeds using high-density EEG and motion capture in 13 healthy participants. The authors report reduced sensorimotor and visual beta power at faster gait speeds, alongside decreased alpha/beta power in the visual cortex during passive viewing of increasing visual flow, suggesting heightened cortical engagement. While this study provides novel insights into multisensory integration during locomotion, methodological concerns arise regarding sample size justification, artifact handling, and cluster validation. The absence of phase-locked analyses and individual response variability further limits mechanistic interpretations. Strengthening statistical rigor and clarifying neurophysiological underpinnings would enhance the work's contribution to understanding gait-related cortical processing. More detailed comments are listed below:   

  1. The introduction cites rodent studies but does not sufficiently bridge these to human locomotion. Please explicitly state how insights from rodent layer 2/3 neurons inform hypotheses about macroscopic EEG signals in humans, given known species differences in movement-related visual responses.
  2. The gamma band is inconsistently defined as 30-50 Hz in Figures S1-S5 but as 30-60 Hz in the Methods. Why?
  3. The use of K-means clustering (k=15) for independent component analysis requires validation. Specify how cluster stability was assessed (e.g., silhouette scores, repeated clustering) and whether dipole locations were anatomically constrained to predefined regions (premotor, sensorimotor, etc.).
  4. Increased parietal alpha power was found during 1.5 m/s walking with faster visual flow (Figure 6), which is attributed to the "overwhelming scale and duration" of mismatches. From my point of view, this explanation is speculative and lacks mechanistic support. Please discuss potential neurophysiological mechanisms (e.g., predictive coding, attention reallocation) and compare with thosefindings in previous studies.
  5. Is visual flow unidirectional, or does it include radial/lamellar components? Since their distinct neural processing, please detail the flow structure and discuss implications for generalizability.
  6. Finally, although mismatched conditions altered electrocortical activity, gait timing remained unchanged. It is necessary to discuss whether cortical adaptations occurred independently of biomechanical outputs or whether the treadmill’s speed constraints masked behavioral effects.

Author Response

Thank you for your detailed comments and invaluable suggestions. Your comments regarding the literature review and the justification of methodology have greatly helped us enhance the logical progression and scientific rigor of our work. We have carefully addressed and incorporated each of your comments into the revised manuscript.  

Comment 1: The introduction cites rodent studies but does not sufficiently bridge these to human locomotion. Please explicitly state how insights from rodent layer 2/3 neurons inform hypotheses about macroscopic EEG signals in humans, given known species differences in movement-related visual responses.

Response 1: We greatly appreciate your suggestion on improving the logical transition and reasoning of our paper. We have revised the introduction accordingly to address this issue (lines 39-44).

Comment 2: The gamma band is inconsistently defined as 30-50 Hz in Figures S1-S5 but as 30-60 Hz in the Methods. Why?

Response 2: Thank you for catching this typographical error. We have corrected the description in the Methods from 30-60 Hz to 30-50 Hz (line 136).

Comment 3: The use of K-means clustering (k=15) for independent component analysis requires validation. Specify how cluster stability was assessed (e.g., silhouette scores, repeated clustering) and whether dipole locations were anatomically constrained to predefined regions (premotor, sensorimotor, etc.).

Response 3: Thank you for your question. We validated the cluster results using repeated clustering and have confirmed the average silhouette score of 0.53 indicating reasonable clustering. Dipole locations were restricted inside the brain of an MNI model, but dipole locations within a cluster were not constrained. The above information has been incorporated into the main text (lines 124-126).

Comment 4: Increased parietal alpha power was found during 1.5 m/s walking with faster visual flow (Figure 6), which is attributed to the "overwhelming scale and duration" of mismatches. From my point of view, this explanation is speculative and lacks mechanistic support. Please discuss potential neurophysiological mechanisms (e.g., predictive coding, attention reallocation) and compare with those findings in previous studies.

Response 4: Thank you for this invaluable feedback. We have expanded the discussion and addressed the potential neurophysiological mechanisms (lines 250-263).

Comment 5: Is visual flow unidirectional, or does it include radial/lamellar components? Since their distinct neural processing, please detail the flow structure and discuss implications for generalizability.

Response 5: Thank you for helping us improve the clarity and depth on this matter. We have provided more details of the flow structure and the suggested discussion in the main text (lines 92-99).

Comment 6: Finally, although mismatched conditions altered electrocortical activity, gait timing remained unchanged. It is necessary to discuss whether cortical adaptations occurred independently of biomechanical outputs or whether the treadmill’s speed constraints masked behavioral effects.

Response 6: We appreciate this invaluable suggestion and have expanded such discussion in the main text (lines 286-291). We also included one more recent reference showing similar results (lines 286-291) suggesting the possibility that the treadmill’s speed constraints had masked gait modifications.

Round 2

Reviewer 2 Report

Comments and Suggestions for Authors

The authors have completed the revisions and adequately addressed my concerns. This manuscript can be accepted in its present form.